# Separable Self-attention for Mobile Vision Transformers

**Sachin Mehta**
*Apple Inc.*

**Mohammad Rastegari**
*Apple Inc.*

**Reviewed on OpenReview:** *https: // openreview. net/ forum? id= tBl4yBEjKi*

## Abstract

Mobile vision transformers (MobileViT) can achieve state-of-the-art performance across several mobile vision tasks, including classification and detection. Though these models have fewer parameters, they have high latency as compared to convolutional neural network-based models. The main efficiency bottleneck in MobileViT is the multi-headed self-attention (MHA) in transformers, which requires $O(k^2)$ time complexity with respect to the number of tokens (or patches) $k$. Moreover, MHA requires costly operations (e.g., batch-wise matrix multiplication) for computing self-attention, impacting latency on resource-constrained devices. This paper introduces a *separable self-attention* method with linear complexity, i.e. $O(k)$. A simple yet effective characteristic of the proposed method is that it uses element-wise operations for computing self-attention, making it a good choice for resource-constrained devices. The improved model, `MobileViTv2`, is state-of-the-art on several mobile vision tasks, including ImageNet object classification and MS-COCO object detection. With about three million parameters, `MobileViTv2` achieves a top-1 accuracy of 75.6% on the ImageNet dataset, outperforming MobileViT by about 1% while running 3.2× faster on a mobile device. Our source code is available at: `https://github.com/apple/ml-cvnets`.

## 1 Introduction

Vision transformers (ViTs) of Dosovitskiy et al. (2021) have become ubiquitous for a wide variety of visual recognition tasks (Touvron et al., 2021; Liu et al., 2021), including mobile vision tasks (Mehta & Rastegari, 2022). At the heart of the ViT-based models, including mobile vision transformers, is the transformer block (Vaswani et al., 2017). The main efficiency bottleneck in ViT-based models, especially for inference on resource-constrained devices, is the multi-headed self-attention (MHA). MHA allows the tokens (or patches) to interact with each other, and is a key for learning global representations. However, the complexity of self-attention in transformer block is $O(k^2)$, i.e., it is quadratic with respect to the number of tokens (or patches) $k$. Besides this, computationally expensive operations (e.g., batch-wise matrix multiplication; see Fig. 1) are required to compute attention matrix in MHA. This, in particular, is concerning for deploying ViT-based models on resource-constrained devices, as these devices have reduced computational capabilities, restrictive memory constraints, and a limited power budget. Therefore, this paper seeks to answer this question: *can self-attention in transformer block be optimized for resource-constrained devices?*

Several methods (e.g., Child et al., 2019; Kitaev et al., 2020; Beltagy et al., 2020; Wang et al., 2020) have been proposed for optimizing the self-attention operation in transformers (not necessarily for ViTs). Among these, a widely studied approach in sequence modeling tasks is to introduce sparsity in self-attention layers, wherein each token attends to a subset of tokens in an input sequence (Child et al., 2019; Beltagy et al., 2020). Though these approaches reduces the time complexity from $O(k^2)$ to $O(k\sqrt{k})$ or $O(k \log k)$, the cost is a performance drop. Another popular approach for approximating self-attention is via low-rank approximation. Linformer (Wang et al., 2020) decomposes the self-attention operation into multiple smaller self-attention

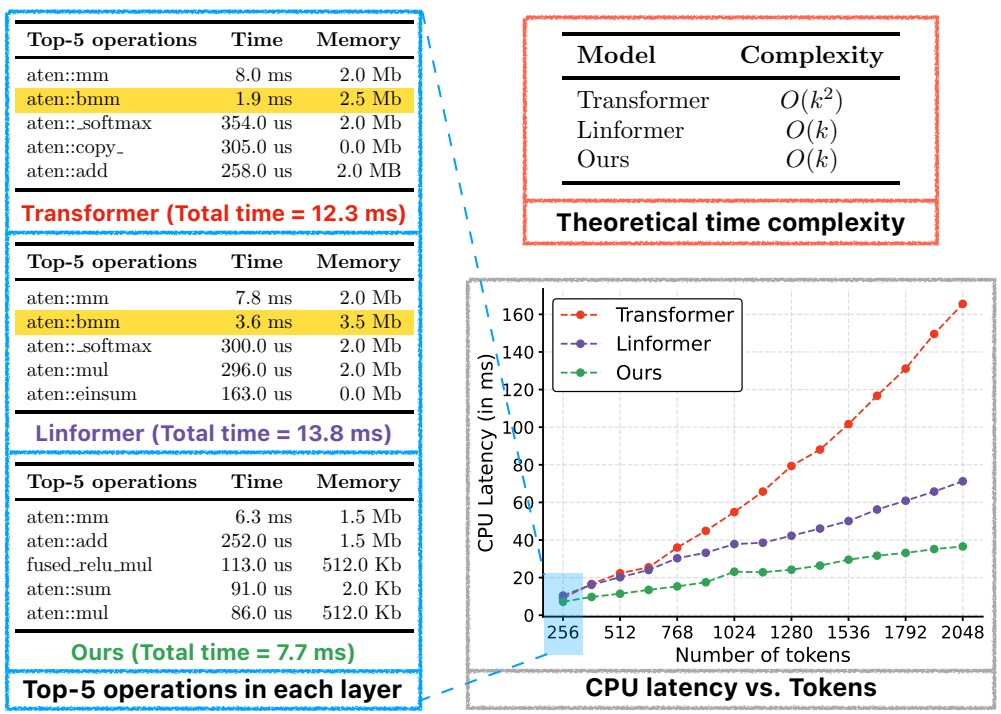

Figure 1: **Comparison between different attention units.** Transformer and Linformer use costly operations (batch-wise matrix multiplication) for computing self-attention. With increase in number of tokens, the cost of computing self-attention (`aten::bmm` operation) outweighs the cost of linear layers; making it the main bottleneck (as also noted in previous works such as Linformer) for efficient inference on resource-constrained devices. The proposed method does not use such operations, thus accelerating inference on resource-constrained devices. **Left** compares top-5 operations (sorted by CPU time) in a single layer of different attention units for $k = 256$ tokens. **Top Right** compares complexity of different attention units. **Bottom Right** compares the latency of different attention units as a function of the number of tokens $k$. These results are computed on a single CPU core machine with a 2.4 GHz 8-Core Intel Core i9 processor, $d = 512$ (token dimensionality), $h = 8$ (number of heads; for Transformer and Linformer), and $p = 256$ (projected tokens in Linformer) using a publicly available profiler in PyTorch Paszke et al. (2019).

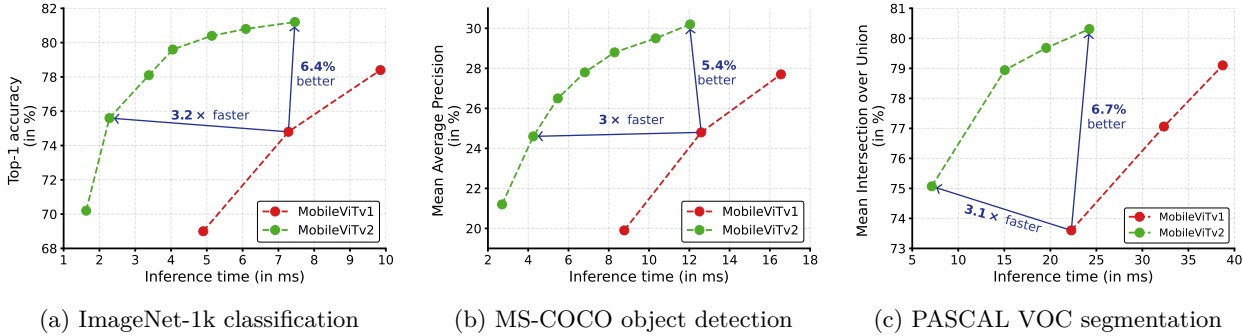

(a) ImageNet-1k classification  (b) MS-COCO object detection  (c) PASCAL VOC segmentation

Figure 2: `MobileViTv2` **models are faster and better than** `MobileViTv1` **models of (Mehta & Rastegari, 2022) across different tasks.** `MobileViTv2` models are constructed by replacing multi-headed self-attention in `MobileViTv1` with the proposed *separable self-attention* (Section 3.2). Here, inference time is measured on an iPhone12 for an input resolution of $256 \times 256$, $512 \times 512$, and $320 \times 320$ for classification, segmentation, and detection respectively.

operations via linear projections, and reduces the complexity of self-attention from $O(k^2)$ to $O(k)$. However, Linformer still uses costly operations (e.g., batch-wise matrix multiplication; Fig. 1) for learning global representations in MHA, which may hinder the deployment of these models on resource-constrained devices.

This paper introduces a novel method, *separable self-attention*, with $O(k)$ complexity for addressing the bottlenecks in MHA in transformers. For efficient inference, the proposed self-attention method also replaces the computationally expensive operations (e.g., batch-wise matrix multiplication) in MHA with element-wise operations (e.g., summation and multiplication). Experimental results on standard vision datasets and tasks demonstrates the effectiveness of the proposed method (Fig. 2).

## 2 Related work

**Improving self-attention**   Improving the efficiency of MHA in transformers is an active area of research. The first line of research introduces locality to address the computational bottleneck in MHA (e.g., Child et al., 2019; Beltagy et al., 2020; Parmar et al., 2018; Qiu et al., 2019; Bello, 2021). Instead of attending to all $k$ tokens, these methods use predefined patterns to limit the receptive field of self-attention from all $k$ tokens to a subset of tokens, reducing the time complexity from $O(k^2)$ to $O(k\sqrt{k})$ or $O(k \log k)$. However, such methods suffer from large performance degradation with moderate training/inference speed-up over the standard MHA in transformers. To improve the efficiency of MHA, the second line of research uses similarity measures to group tokens (Kitaev et al., 2020; Vyas et al., 2020; Wang et al., 2021a). For instance, Reformer (Kitaev et al., 2020) uses locality-sensitive hashing to group the tokens and reduces the theoretical self-attention cost from $O(k^2)$ to $O(k \log k)$. However, the efficiency gains over standard MHA are noticeable only for large sequences ($k > 2048$) (Kitaev et al., 2020). Because $k < 1024$ in ViTs, these approaches are not suitable for ViTs. The third line of research improves the efficiency of MHA via low-rank approximation (Wang et al., 2020; Choromanski et al., 2020). The main idea is to approximate the self-attention matrix with a low-rank matrix, reducing the computational cost from $O(k^2)$ to $O(k)$. Even though these methods speed-up the self-attention operation significantly, they still use expensive operations for computing attention, which may hinder the deployment of these models on resource-constrained devices (Fig. 1).

In summary, existing methods for improving MHA are limited in their reduction of inference time and memory consumption, especially for resource-constrained devices. This work introduces a separable self-attention method that is fast and memory-efficient (see Fig. 1), which is desirable for resource-constrained devices.

**Improving transformer-based models**   There has been significant work on improving the efficiency of transformers (Liu et al., 2021; Mehta & Rastegari, 2022; Mehta et al., 2021; Wu et al., 2021; Heo et al., 2021). The majority of these approaches reduce the number of tokens in the transformer block using different methods, including down-sampling (Ryoo et al., 2021; Heo et al., 2021) and pyramidal structure (Liu et al., 2021; Wang et al., 2021b; Mehta & Rastegari, 2022). Because the proposed separable self-attention module is a drop-in replacement to MHA, it can be easily integrated with any transformer-based model to further improve its efficiency.

**Other methods**   Transformer-based models performance can be improved using different methods, including mixed-precision training (Micikevicius et al., 2018), efficient optimizers (Dettmers et al., 2022; Zhai et al., 2021a), and knowledge distillation (Touvron et al., 2021; Kundu & Sundaresan, 2021). These methods are orthogonal to our work, and by default, we use mixed-precision during training.

## 3 MobileViTv2

MobileViT of Mehta & Rastegari (2022) is a hybrid network that combines the strengths of CNNs and ViTs. MobileViT views transformers as convolutions, which allows it to leverage the merits of both convolutions (e.g., inductive biases) and transformers (e.g., long-range dependencies) to build a light-weight network for mobile devices. Though MobileViT networks have significantly fewer parameters and deliver better performance as compared to light-weight CNNs (e.g., MobileNets (Sandler et al., 2018; Howard et al., 2019)), they have high latency. The main efficiency bottleneck in MobileViT is the multi-headed self-attention (MHA; Fig. 3a).

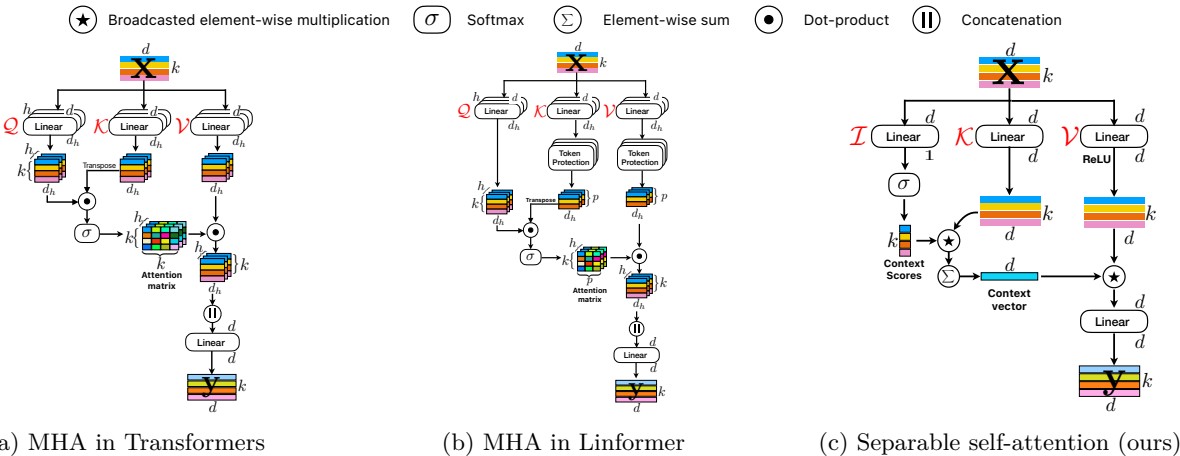

(a) MHA in Transformers      (b) MHA in Linformer      (c) Separable self-attention (ours)

Figure 3: **Different self-attention units.** **(a)** is a standard multi-headed self-attention (MHA) in transformers. **(b)** extends MHA in (a) by introducing token projection layers, which project $k$ tokens to a pre-defined number of tokens $p$, thus reducing the complexity from $O(k^2)$ to $O(k)$. However, it still uses costly operations (e.g., batch-wise matrix multiplication) for computing self-attention, impacting latency on resource-constrained devices (Fig. 1). **(c)** is the proposed separable self-attention layer that is linear in complexity, i.e., $O(k)$, and uses element-wise operations for faster inference.

MHA uses scaled dot-product attention to capture the contextual relationships between $k$ tokens (or patches). However, MHA is expensive as it has $O(k^2)$ time complexity. This quadratic cost is a bottleneck for transformers with a large number of tokens $k$ (Fig. 1). Moreover, MHA uses computationally- and memory-intensive operations (e.g., batch-wise matrix multiplication and softmax for computing attention matrix; Fig. 1); which could be a bottleneck on resource-constrained devices. To address the limitations of MHA for efficient inference on resource-constrained devices, this paper introduces *separable self-attention* with linear complexity (Fig. 3c).

The main idea of our *separable self-attention* approach, shown in Fig. 4b, is to compute context scores with respect to a latent token $L$. These scores are then used to re-weight the input tokens and produce a context vector, which encodes the global information. Because the self-attention is computed with respect to a latent token, the proposed method can reduce the complexity of self-attention in the transformer by a factor $k$. A simple yet effective characteristic of the proposed method is that it uses element-wise operations (e.g., summation and multiplication) for its implementation, making it a good choice for resource-constrained devices. We call the proposed attention method separable self-attention because it allows us to encode global information by replacing the quadratic MHA with two separate linear computations. The improved model, `MobileViTv2`, is obtained by replacing MHA with separable self-attention in MobileViT.

In the rest of this section, we first briefly describe MHA (Section 3.1), and then elaborate on the details of separable self-attention (Section 3.2) and `MobileViTv2` architecture (Section 3.3).

### 3.1 Overview of multi-headed self-attention

MHA (Fig. 3a) allows transformer to encode inter-token relationships. Specifically, MHA takes an input $\mathbf{x} \in \mathbb{R}^{k \times d}$ comprising of $k$ $d$-dimensional token (or patch) embeddings. The input $\mathbf{x}$ is then fed to three branches, namely query $\mathcal{Q}$, key $\mathcal{K}$, and value $\mathcal{V}$. Each branch ($\mathcal{Q}$, $\mathcal{K}$, and $\mathcal{V}$) is comprised of $h$ linear layers (or heads), which enables the transformer to learn multiple views of the input. The dot-product between the output of linear layers in $\mathcal{Q}$ and $\mathcal{K}$ is then computed simultaneously for all $h$ heads, and is followed by a softmax operation $\sigma$ to produce an attention (or context-mapping) matrix $\mathbf{a} \in \mathbb{R}^{k \times k \times h}$. Another dot-product is then computed between $\mathbf{a}$ and the output of linear layers in $\mathcal{V}$ to produce weighted sum output $\mathbf{y_w} \in \mathbb{R}^{k \times d_h \times h}$, where $d_h = \frac{d}{h}$ is the head dimension . The outputs of $h$ heads are concatenated to produce a tensor with $k$ $d$-dimensional tokens, which is then fed to another linear layer with weights $\mathbf{W_O} \in \mathbb{R}^{d \times d}$ to

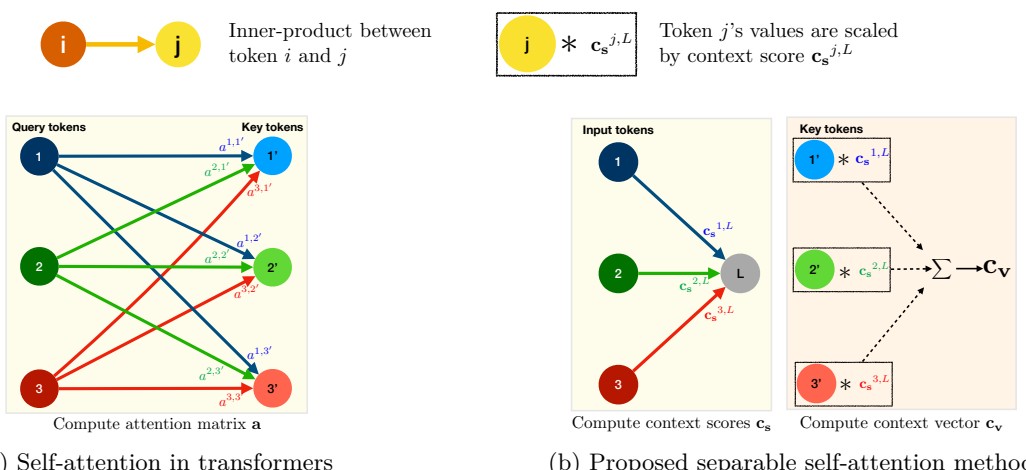

Figure 4: **Example illustrating the interaction between tokens to learn global representations in different attention layers.** In (a), each query token computes the distance with all key tokens via dot-product. These distances are then normalized using softmax to produce an attention matrix **a**, which encodes contextual relationships. In (b), the inner product between input tokens and latent token $L$ is computed. The resultant vector is normalized using softmax to produce context scores $\mathbf{c_s}$. These context scores are used to weight key tokens and produce a context vector $\mathbf{c_v}$, which encodes contextual information.

produce the output of MHA $\mathbf{y} \in \mathbb{R}^{k \times d}$. Mathematically, this operation can be described as:

$$\mathbf{y} = \text{Concat} \left( \langle \underbrace{\sigma \left( \langle \mathbf{xW_Q}^0, \mathbf{xW_K}^0 \rangle \right)}_{\mathbf{a}^0 \in \mathbb{R}^{k \times k}}, \mathbf{xW_V}^0 \rangle, \cdots, \langle \underbrace{\sigma \left( \langle \mathbf{xW_Q}^h, \mathbf{xW_K}^h \rangle \right)}_{\mathbf{a}^h \in \mathbb{R}^{k \times k}}, \mathbf{xW_V}^h \rangle \right) \mathbf{W_O} \tag{1}$$

where $\mathbf{W_Q}^i \in \mathbb{R}^{d \times d_h}$, $\mathbf{W_K}^i \in \mathbb{R}^{d \times d_h}$, and $\mathbf{W_V}^i \in \mathbb{R}^{d \times d_h}$ are the weights of the $i$-th linear layer (or head) in $\mathcal{Q}$, $\mathcal{K}$, and $\mathcal{V}$ branches respectively. The symbol $\langle \cdot, \cdot \rangle$ denotes the dot-product operation.

## 3.2 Separable self-attention

The structure of separable self-attention is inspired by MHA. Similar to MHA, the input $\mathbf{x}$ is processed using three branches, i.e., input $\mathcal{I}$, key $\mathcal{K}$, and value $\mathcal{V}$. The input branch $\mathcal{I}$ maps each $d$-dimensional token in $\mathbf{x}$ to a scalar using a linear layer with weights $\mathbf{W_I} \in \mathbb{R}^d$. The weights $\mathbf{W_I}$ serves as the latent node $L$ in Fig. 4b. This linear projection is an inner-product operation and computes the distance between latent token $L$ and $\mathbf{x}$, resulting in a $k$-dimensional vector. A softmax operation is then applied to this $k$-dimensional vector to produce context scores $\mathbf{c_s} \in \mathbb{R}^k$. Unlike transformers that compute the attention (or context) score for each token with respect to all $k$ tokens, the proposed method only computes the context score with respect to a latent token $L$. This reduces the cost of computing attention (or context) scores from $O(k^2)$ to $O(k)$.

The context scores $\mathbf{c_s}$ are used to compute a context vector $\mathbf{c_v}$. Specifically, the input $\mathbf{x}$ is linearly projected to a $d$-dimensional space using key branch $\mathcal{K}$ with weights $\mathbf{W_K} \in \mathbb{R}^{d \times d}$ to produce an output $\mathbf{x_K} \in \mathbb{R}^{k \times d}$. The context vector $\mathbf{c_v} \in \mathbb{R}^d$ is then computed as a weighted sum of $\mathbf{x_K}$ as:

$$\mathbf{c_v} = \sum_{i=1}^{k} \mathbf{c_s}(i) \mathbf{x_K}(i) \tag{2}$$

The context vector $\mathbf{c_v}$ is analogous to the attention matrix **a** in Eq. (1) in a sense that it also encodes the information from all tokens in the input $\mathbf{x}$, but is cheap to compute.

The contextual information encoded in $\mathbf{c_v}$ is shared with all tokens in $\mathbf{x}$. To do so, the input $\mathbf{x}$ is linearly projected to a $d$-dimensional space using a value branch $\mathcal{V}$ with weights $\mathbf{W_V} \in \mathbb{R}^{d \times d}$, followed by a ReLU

Table 1: **Effect of different self-attention methods** on the performance of MobileViT on the ImageNet-1k dataset. Here, all models have similar number of parameters and FLOPs, and latency is measured on iPhone12 (CPU and neural engine (NE)).

| Attention unit | Latency ↓ | | Top-1 ↑ |
|---|---|---|---|
| | **NE** | **CPU** | |
| Self-attention in Transformer (Fig. 3a) | 9.9 ms | 78.6 ms | **78.4** |
| Self-attention in Linformer (Fig. 3b) | 10.2 ms | 83.2 ms | 78.2 |
| Separable self-attention (Ours; Fig. 3c) | **3.4** ms | **38.6** ms | 78.1 |

activation to produce an output $\mathbf{x_V} \in \mathbb{R}^{k \times d}$. The contextual information in $\mathbf{c_v}$ is then propagated to $\mathbf{x_V}$ via broadcasted element-wise multiplication operation. The resultant output is then fed to another linear layer with weights $\mathbf{W_O} \in \mathbb{R}^{d \times d}$ to produce the final output $\mathbf{y} \in \mathbb{R}^{k \times d}$. Mathematically, separable self-attention can be defined as:

$$\mathbf{y} = \left( \underbrace{\sum \left( \overbrace{\sigma\left(\mathbf{xW_I}\right)}^{\mathbf{c_s} \in \mathbb{R}^k} * \mathbf{xW_K} \right)}_{\mathbf{c_v} \in \mathbb{R}^d} * \mathrm{ReLU}\left(\mathbf{xW_V}\right) \right) \mathbf{W_O} \tag{3}$$

where $*$ and $\sum$ are broadcastable element-wise multiplication and summation operations, respectively.

**Comparison with self-attention methods** Fig. 1 compares the proposed method with Transformer and Linformer[1]. Because time complexity of self-attention methods do not account for the cost of operations that are used to implement these methods, some of the operations may become bottleneck on resource-constrained devices. For holistic understanding, module-level latency on a single CPU core with varying $k$ is also measured in addition to theoretical metrics. The proposed separable self-attention is fast and efficient as compared to MHA in Transformer and Linformer.

Besides these module-level results, when we replaced the MHA in the transformer with the proposed self-separable attention in the MobileViT architecture, we observe $3\times$ improvement in inference speed with similar performance on the ImageNet-1k dataset (Table 1). These results show the efficacy of the proposed separable self-attention at the architecture-level. Note that self-attention in Transformer and Linformer yields similar results for MobileViT. This is because the number of tokens $k$ in MobileViT is fewer ($k \leq 1024$) as compared to language models, where Linformer is significantly faster than the transformer.

**Relationship with additive addition** The proposed approach resembles the attention mechanism of Bahdanau et al. (2014), which also encodes the global information by taking a weighted-sum of LSTM outputs at each time step. Unlike Bahdanau et al. (2014), where input tokens interact via recurrence, the input tokens in the proposed method interact only with a latent token.

### 3.3 `MobileViTv2` architecture

To demonstrate the effectiveness of the proposed separable self-attention on resource-constrained devices, we integrate separable self-attention with a recent ViT-based model, MobileViT (Mehta & Rastegari, 2022). MobileViT is a light-weight, mobile-friendly hybrid network that delivers significantly better performance than other competitive CNN-based, transformer-based, or hybrid models, including MobileNets (Howard et al.,

---

[1]Inspired by sampling-based methods (e.g., DynamicViT (Rao et al., 2021)) for reducing the computational cost of self-attention recent works, we also studied the proposed method with different sampling methods (random, uniform, top-k). We observe that these approaches are effective in reducing the FLOPs of the model with little or no drop in performance. However, these sampling-based methods have high latency on mobile devices as compared to the models without these methods (see Appendix E). This is likely because these approaches have high memory access cost (because of tensor memory re-ordering) and constitutes to large portion of latency.

2017; Sandler et al., 2018; Howard et al., 2019). To avoid ambiguity, we refer to MobileViT as MobileViTv1 in the rest of the paper.

Specifically, we replace MHA in the transformer block in the MobileViTv1 with the proposed separable self-attention method. We call the resultant architecture `MobileViTv2`. We also do not use the skip-connection and fusion block in the MobileViT block (Fig. 1b in (Mehta & Rastegari, 2022)) as it improves the performance marginally (Fig. 12 in (Mehta & Rastegari, 2022)). Furthermore, to create `MobileViTv2` models at different complexities, we uniformly scale the width of `MobileViTv2` network using a width multiplier $\alpha \in \{0.5, 2.0\}$. This is in contrast to MobileViTv1 which trains three specific architectures (XXS, XS, and S) for mobile devices. More details about `MobileViTv2`'s architecture are given in Appendix A.

## 4 Experimental results

### 4.1 Object classification on the ImageNet dataset

**Training on ImageNet-1k from scratch**  We train `MobileViTv2` for 300 epochs with an effective batch size of 1024 images (128 images per GPU × 8 GPUs) using AdamW of Loshchilov & Hutter (2019) on the ImageNet-1k dataset (Russakovsky et al., 2015) with 1.28 million and 50 thousand training and validation images respectively. We linearly increase the learning rate from $10^{-6}$ to 0.002 for the first 20k iterations. After that, the learning rate is decayed using a cosine annealing policy (Loshchilov & Hutter, 2017). To reduce stochastic noise during training, we use exponential moving average (EMA) (Polyak & Juditsky, 1992) as we find it helps larger models. We implement our models using `CVNets` (Mehta & Rastegari, 2022; Mehta et al., 2022), and use their provided scripts for data processing, training, and evaluation.

**Pre-training on ImageNet-21k-P and finetuning on ImageNet-1k**  We train on the ImageNet-21k (winter'21 release) that contains about 13 million images across 19k classes. Specifically, we follow Ridnik et al. (2021) to pre-process (e.g., remove classes with fewer samples) the dataset and split it into about 11 million and 522 thousand training and validation images spanning over 10,450 classes, respectively. Following Ridnik et al. (2021), we refer to this pre-processed dataset as ImageNet-21k-P. Note that the ImageNet-21k-P validation set does not overlap with the validation and test sets of ImageNet-1k.

We follow Ridnik et al. (2021) for pre-training `MobileViTv2` on ImageNet-21k-P. For faster convergence, we initialize `MobileViTv2` models with ImageNet-1k weights and finetune it on ImageNet-21k-P for 80 epochs with an effective batch size of 4096 images (128 images per GPU x 32 GPUs). We do not use any linear warm-up. Other settings follow ImageNet-1k training.

We finetune ImageNet-21k-P pre-trained models on ImageNet-1k for 50 epochs using SGD with momentum (0.9) and cosine annealing policy with an effective batch size of 256 images (128 images per GPU × 2 GPUs).

**Finetuning at higher resolution**  `MobileViTv2` is a hybrid architecture that combines convolution and separable self-attention to learn visual representations. Unlike many ViT-based models (e.g., DeiT), `MobileViTv2` does not require adjustment to patch embeddings or positional biases for different input resolutions and is simple to finetune. We finetune `MobileViTv2` models at higher resolution (i.e., 384×384) for 10 epochs with a fixed learning rate of $10^{-3}$ using SGD.

**Comparison with existing methods**  Table 2 and Fig. 2 compares `MobileViTv2`'s performance with recent methods. We make following observations:

- When MHA in MobileViTv1 is replaced with separable self-attention, the resultant model, `MobileViTv2`, is faster and better (Fig. 2); validating the effectiveness of the proposed separable self-attention method for mobile ViTs.

- Compared to transformer-based (including hybrid) models, `MobileViTv2` models are fast on mobile devices. For example, `MobileViTv2` is about 8× faster on a mobile device and delivers 2.5% better performance on the ImageNet-1k dataset than MobileFormer (Chen et al., 2021b), even though MobileFormer is FLOP efficient (R9 vs. R10). However, on GPU, both MobileFormer and `MobileViTv2` run at a similar

Table 2: **Classification performance on the ImageNet-1k validation set**. Here, NS means that we are not able to measure the latency on mobile device as some operations (e.g., cyclic shifts) are not supported on mobile devices. Following Mehta & Rastegari (2022), latency is measured on iPhone12 with a batch size of 1. Similar to Liu et al. (2021) and Liu et al. (2022), throughput is measured on NVIDIA V100 GPUs with a batch size of 128. The rows are grouped by network parameters.

| Row # | Model | Type | Neural search? | Extra data | Image size | # Params ↓ | FLOPs ↓ | Latency ↓ (in ms) | Throughput ↑ (images/ sec) | Top-1 ↑ (in %) |
|---|---|---|---|---|---|---|---|---|---|---|
| R1 | MobileViT-XXS | Hybrid | ✗ | None | $256^2$ | **1.3 M** | 0.4 G | 4.8 | 4225 | 69.0 |
| R2 | MobileViTv2-0.5 | Hybrid | ✗ | None | $256^2$ | 1.4 M | 0.5 G | **1.6** | **4595** | **70.2** |
| R3 | MobileFormer-52 | Hybrid | ✗ | None | $224^2$ | 3.6 M | **52 M** | 7.1 | 4445 | 68.7 |
| R4 | MobileViTv2-1.0 | Hybrid | ✗ | None | $256^2$ | **4.9 M** | 1.8 G | 3.4 | 2351 | 78.1 |
| R5 | EfficientNet-b0 | CNN | ✓ | None | $224^2$ | 5.3 M | **422 M** | **1.6** | **4619** | 77.1 |
| R6 | DeiT-Tiny | Transformer | ✗ | None | $224^2$ | 5.5 M | 1.3 G | 3.4 | 4541 | 72.2 |
| R7 | MobileViT-S | Hybrid | ✗ | None | $256^2$ | 5.6 M | 2.0 G | 9.9 | 1986 | **78.4** |
| R8 | EfficientNet-b2 | CNN | ✓ | None | $288^2$ | **9.1 M** | 1.2 G | **3.8** | **2032** | 80.1 |
| R9 | MobileViTv2-1.5 | Hybrid | ✗ | None | $256^2$ | 10.6 M | 4.0 G | 5.1 | 1418 | **80.4** |
| R10 | MobileFormer-294 | Hybrid | ✗ | None | $224^2$ | 11.8 M | **294 M** | 40.7 | 1402 | 77.9 |
| R11 | MobileViTv2-2.0 | Hybrid | ✗ | None | $256^2$ | **18.5 M** | 7.5 G | 7.5 | 1105 | 81.2 |
| R12 | Swin-T | Hybrid | ✗ | None | $224^2$ | 28.3 M | **4.5 G** | NS | 1390 | 81.3 |
| R13 | ConvNext-T | CNN | ✗ | None | $224^2$ | 28.6 M | **4.5 G** | **3.7** | **1800** | 82.1 |
| R14 | DeiT-Base | Transformer | ✗ | None | $224^2$ | 86.6 M | 17.6 G | 13.2 | 958 | 81.8 |
| R15 | MobileViTv2-2.0 | Hybrid | ✗ | ImageNet-21k-P | $256^2$ | **18.5 M** | 7.5 G | 7.5 | 1105 | 82.4 |
| R16 | ConvNext-T | CNN | ✗ | ImageNet-21k | $224^2$ | 28.6 M | **4.5 G** | **3.7** | **1800** | **82.9** |
| R17 | MobileViTv2-2.0 | Hybrid | ✗ | ImageNet-21k-P | $384^2$ | **18.5 M** | 16.1 G | 17.0 | 488 | 83.4 |
| R18 | ConvNext-T | CNN | ✗ | ImageNet-21k | $384^2$ | 28.6 M | **13.1 G** | **8.6** | **645** | **84.1** |

speed. The discrepancy in FLOPs and speed of MobileFormer across devices is primarily because of its architectural design. MobileFormer has conditional operations between mobile and former blocks. Such conditional operations, especially on resource-constrained devices, have a low degree of parallelism and create memory bottlenecks, resulting in a high latency network. Ma et al. (2018) also makes a similar observation for CNN-based architectures.

- `MobileViTv2` bridges the latency gap between CNN- and ViT-based models on mobile devices while maintaining performance with similar or fewer parameters. For example, on a mobile device, ConvNexT (Liu et al., 2022) (CNN-based model) is 2× and 3.6× faster than `MobileViTv2` (hybrid model) and DeiT (transformer-based model) for similar performance respectively (see R11, R13, and R14). The low latency of fully CNN-based models on mobile devices can be attributed to several device-level optimizations that have been done for CNN-based models over the past few years (e.g., dedicated hardware implementations for convolutions and folding batch normalization with convolutions). ViT-based models still lack such optimizations and therefore, the resultant inference graphs are sub-optimal. Though `MobileViTv2` bridges the latency gap between CNNs and ViTs, we believe the latency of ViT-based models will improve in the future with similar optimizations.

- The delta in speed (on GPU) between ConvNext and `MobileViTv2` (R15-R18) at higher model complexities reduces from 1.6× to 1.3× when input resolution is increased from 224 × 224 (or 256 × 256) to 384 × 384, suggesting ViT-based (including hybrid) models exhibit better scaling properties as compared to CNNs. This is because of a higher degree of parallelism that ViT-based models offer at a large scale (Dosovitskiy et al., 2021; Brown et al., 2020). Our results on down-stream tasks in Section 4.2 and previous work on scaling ViTs (Dosovitskiy et al., 2021; Zhai et al., 2021b) further supports this observation.

See Appendix for more results on the task of image classification (e.g., Appendix B for `MobileViTv2`'s classification performance on ImageNet-1k/21k-P, Appendix C for comparison with light-weight models, and Appendix E for ablations, including the effect of augmentation, loss functions, multiple latent tokens, and locality-based sampling methods on FLOP efficiency).

Table 3: **Semantic segmentation results on the ADE20k and the PASCAL VOC 2012 datasets.** Here, throughput, network parameters, and FLOPs are measured on the ADE20k dataset for an input with a spatial resolution of $512 \times 512$. mIoU (mean intersection over union) score is calculated for a single scale only. Throughput is calculated using a batch size of 32 images on a single NVIDIA V100 GPU with 32 GB memory and is an average of over 50 iterations (excluding 10 iterations for warmup). We do not report latency on a mobile device as some of the operations (e.g., pyramid pooling in PSPNet) are not optimally implemented for mobile devices. The baseline results are from MMSegmentation (2020). Rows are grouped by network parameters.

| Seg. Model | ImageNet-1k Backbone | Image Size | Throughput ↑ (images/sec) | # Params ↓ (in millions) | FLOPs ↓ (in billions) | mIoU ↑ | |
|---|---|---|---|---|---|---|---|
| | | | | | | ADE20k | PASCAL VOC |
| PSPNet | MobileViTv2-0.5 (Ours) | $512^2$ | **439** | **3.6 M** | **15.4 G** | **31.8** | 74.6 |
| | MobileNetv2 | $512^2$ | 276 | 13.7 M | 53.1 G | 29.7 | – |
| PSPNet | MobileViTv2-1.75 (Ours) | $512^2$ | 114 | **22.5 M** | **95.9 G** | 39.8 | **80.2** |
| | ResNet-50 | $512^2$ | **119** | 49.1 M | 179.1 G | **41.1** | 76.8 |
| DeepLabv3 | MobileViTv2-0.75 (Ours) | $512^2$ | 241 | **9.6 M** | **40.0 G** | **34.7** | 75.1 ($\alpha$=0.5) |
| | MobileNetv2 | $512^2$ | **246** | 18.7 M | 75.4 G | 34.1 | – |
| DeepLabv3 | MobileViTv2-2.0 (Ours) | $512^2$ | 90 | **34.0 M** | **147.0 G** | 40.9 | **80.3** ($\alpha$=1.5) |
| | ResNet-50 | $512^2$ | **103** | 68.2 M | 270.3 G | **42.4** | 79.1 |

## 4.2 Evaluation on down-stream tasks

**Semantic segmentation** We integrate `MobileViTv2` with two standard segmentation architectures, PSPNet (Zhao et al., 2017) and DeepLabv3 (Chen et al., 2017), and study it on two standard semantic segmentation datasets, ADE20k (Zhou et al., 2017) and PASCAL VOC 2012 (Everingham et al., 2015). For training details including hyper-parameters, see supplementary material.

Table 3 and Fig. 2c compares the segmentation performance in terms of validation mean intersection over union (mIOU) of `MobileViTv2` with different segmentation methods. `MobileViTv2` delivers competitive performance at different complexities while having significantly fewer parameters and FLOPs. Interestingly, the inference speed of `MobileViTv2` models is comparable to CNN-based models, including light-weight MobileNetv2 and heavy-weight ResNet-50 (He et al., 2016) model. This is consistent with our observation in Section 4.1 (R17 vs. R18; Table 2) where we also observe that ViT-based models scale better than CNN's at higher input resolutions and model complexities.

**Object detection** We integrate `MobileViTv2` with SSDLite of Sandler et al. (2018) (SSD head (Liu et al., 2016) with separable convolutions) for mobile object detection, and study its performance on MS-COCO dataset (Lin et al., 2014). We follow Mehta & Rastegari (2022) for training detection models. Table 4 and Fig. 2b compares SSDLite's detection performance in terms of validation mean average precision (mAP) using different ImageNet-1k backbones. `MobileViTv2` delivers competitive performance to models with different capacities, further validating the effectiveness of the proposed self-separable attention method.

## 5 Visualizations of self-separable attention scores

Fig. 5 visualizes what the context scores learn at different output strides[2] of `MobileViTv2` network. We found that separable self-attention layers pay attention to low-, mid-, and high-level features, and allow `MobileViTv2` to learn representations from semantically relevant image regions.

## 6 Robustness analysis of self-separable attention layer

Previous works have shown that vision transformer (ViTs) models generalizes better and robust as compared to CNNs. A natural question arises is: Do separable self-attention layer exhibit similar properties? To study the robustness of `MobileViTv2`, we evaluate the performance of `MobileViTv2` on the ImageNetv2 dataset

---

[2]Output stride is the ratio of the spatial dimension of the input to the feature map.

Table 4: **Object detection using SSDLite on the MS-COCO dataset.** Here, throughput is measured with a batch of 128 images on the NVIDIA V100 GPU, and is an average over 50 iterations (excluding 10 iterations for warmup). Latency on a mobile device is not reported as some operations (e.g., hard swish) are not optimally implemented for such devices. Rows are grouped by network parameters.

| ImageNet-1k backbone | Image Size | Throughput ↑ (images/sec) | # Params ↓ (in millions) | FLOPs ↓ (in billions) | mAP ↑ |
|---|---|---|---|---|---|
| MobileViTv1-XXS | $320^2$ | 2246 | **1.7 M** | 0.9 G | 19.9 |
| MobileViTv2-0.5 (Ours) | $320^2$ | 2782 | 2.0 M | 0.9 G | 21.2 |
| MobileViTv2-0.75 (Ours) | $320^2$ | 1876 | 3.6 M | 1.8 G | **24.6** |
| Mobilenetv2 | $320^2$ | 3052 | 4.3 M | 0.8 G | 22.1 |
| MobileNetv3 | $320^2$ | 3884 | 5.0 M | **0.6 G** | 22.0 |
| MobileNetv1 | $320^2$ | **4330** | 5.1 M | 1.3 G | 22.2 |
| MobileViTv2-1.75 | $320^2$ | 780 | **14.9 M** | **9.0 G** | **29.5** |
| ResNet-50 | $300^2$ | 744 | 22.9 M | 20.2 G | 25.2 |

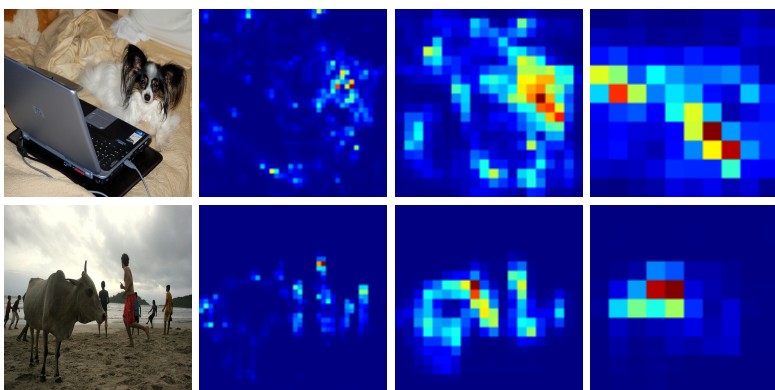

Figure 5: **Context score maps at different output strides (OS) of `MobileViTv2` model.** Observe how context scores pay attention to semantically relevant image regions. (**Left to right:** input image, context scores at OS=8, context scores at OS=16, and context scores at OS=32). For more examples and details about context score map generation, see Appendix D.

| Model | # Params ↓ | ImageNet Top-1 ↑ | ImageNetv2 Top-1 ↑ |
|---|---|---|---|
| DeiT-T | 5 M | 72.2 | 60.4 |
| MobileViT-S | 5.5 M | **78.4** | **66.5** |
| MobileViTv2-1.5 | **4.9 M** | 78.1 | 66.4 |
| DeiT-S | 22 M | 79.8 | 68.5 |
| MobileViTv2-2.0 | **18.5 M** | **81.2** | **70.0** |

Table 5: **`MobileViTv2` models are robust and generalizes to harder images.**

(matched frequency split) (Recht et al., 2019). Table 5 shows that `MobileViTv2` models exhibit similar robustness properties to transformer-based models (DeiT and MobileViT) and generalizes to harder images.

# 7 Conclusions

Transformer-based vision models are slow on mobile devices as compared to CNN-based models because multi-headed self-attention is expensive on resource-constrained devices. In this paper, we introduce a *separable self-attention* method that has linear complexity and can be implemented using hardware-friendly

element-wise operations. Experimental results on standard datasets and tasks demonstrate the effectiveness of the proposed method over multi-headed self-attention.

In this paper, we do not use methods (e.g., distillation and neural architecture search), which have found to be effective in improving the accuracy and efficiency of recent efficient vision transformer models (Li et al., 2022). In the future, we plan to study such methods to further improve the accuracy and efficiency of `MobileViTv2` models.

## Acknowledgements

We are grateful to Ali Farhadi, Peter Zatloukal, Oncel Tuzel, Rick Chang, Fartash Faghri, Farzad Abdolhosseini, Lailin Chen, and Max Horton for their helpful comments. We are also thankful to Apple's infrastructure and open-source teams for their help with training infrastructure and open-source release of the code and pre-trained models.

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

## A   Detailed architecture of `MobileViTv2`

`MobileViTv2`'s architecture follows MobileViTv1 (Mehta & Rastegari, 2022) and is given in Table 6. `MobileViTv2` block, shown in Fig. 6, makes two changes to the MobileViTv1 block: (1) it replaces the multi-headed self-attention with the proposed separable self-attention to learn global representations and (2) it does not use fusion block and skip-connection (see Fig. 1b in Mehta & Rastegari (2022)) as they improve the performance marginally (see Fig. 12 in Mehta & Rastegari (2022)). The expansion factor in MobileNetv2 (Sandler et al., 2018) blocks and feed-forward layers is two. Similar to Mehta & Rastegari (2022), we use Swish (Elfwing et al., 2018) as a non-linear activation function. Unlike MobileViTv1 that creates three specific architectures (XXS, XS, and S) for mobile devices, we uniformly scale the width of `MobileViTv2` network using a width multiplier $\alpha \in 0.5, 2.0$ to create models at different complexities.

## B   `MobileViTv2`'s classification performance

**ImageNet-1k**   Table 7 shows the results of `MobileViTv2` on the ImageNet-1k dataset. Finetuning `MobileViTv2` models at higher resolution ($384 \times 384$) shows improvement across the board. For example, the performance of `MobileViTv2`-0.50 with 1.4 million parameters improves by about 2% when finetuned at higher resolution (R1 vs. R2). Similarly, pre-training on the ImageNet-21k-P dataset helps improve the performance of `MobileViTv2` models. For example, ImageNet-21k-P pretraining improves the performance of `MobileViTv2`-2.0 improves by 1.2% (R17 vs. R18). Notably, `MobileViTv2` models pretrained on the

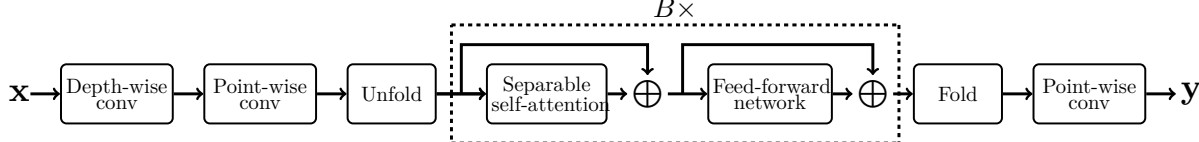

Figure 6: **`MobileViTv2` block**. Here, depth-wise convolution uses a kernel size of $3 \times 3$ to encode local representations. Similar to MobileViTv1, unfolding and folding operations uses a patch height and width of two respectively. The separable self-attention and feed-forward layers are repeated $B\times$ before applying the folding operation.

Table 6: **`MobileViTv2` architecture.**   Here, $d$ represents dimensionality of the input to the separable self-attention layer, $B$ denotes the repetition of transformer block with separable self-attention inside the `MobileViTv2` block (Fig. 6), and MV2 indicates MobileNetv2 block. Similar to MobileViTv1 block, we set kernel size as three and spatial dimensions of patch (height $h$ and width $w$) as two in the `MobileViTv2` block.

| Layer | Output size | Output stride | Repeat | Output channels |
|---|---|---|---|---|
| Image | $256 \times 256$ | 1 | | |
| Conv-$3 \times 3$, $\downarrow 2$
MV2 | $128 \times 128$ | 2 | 1
1 | $32\alpha$
$64\alpha$ |
| MV2, $\downarrow 2$
MV2 | $64 \times 64$ | 4 | 1
2 | $128\alpha$
$128\alpha$ |
| MV2, $\downarrow 2$
`MobileViTv2` block (Fig. 6; $B = 2$) | $32 \times 32$ | 8 | 1
1 | $256\alpha$
$256 * \alpha$ $(d = 128\alpha)$ |
| MV2, $\downarrow 2$
`MobileViTv2` block (Fig. 6; $B = 4$) | $16 \times 16$ | 16 | 1
1 | $384\alpha$
$384\alpha$ $(d = 192\alpha)$ |
| MV2, $\downarrow 2$
`MobileViTv2` block (Fig. 6; $B = 3$) | $8 \times 8$ | 32 | 1
1 | $512\alpha$
$512\alpha$ $(d = 256\alpha)$ |
| Global pool
Linear | $1 \times 1$ | 256 | 1 | $512\alpha$
1000 |

Table 7: **Classification performance of `MobileViTv2` on the ImageNet-1k dataset.** Here, $^\dagger$ indicates finetuning at higher resolution.

| Row # | Model | Image size | Extra data | # Params ↓ | FLOPs ↓ | Top-1 ↑ |
|---|---|---|---|---|---|---|
| R1 | MobileViTv2-0.50 | $256^2$ | None | 1.4 M | 0.5 G | 70.2 |
| R2 | MobileViTv2-0.50$^\dagger$ | $384^2$ | None | 1.4 M | 1.0 G | 72.1 |
| R3 | MobileViTv2-0.75 | $256^2$ | None | 2.9 M | 1.0 G | 75.6 |
| R4 | MobileViTv2-0.75$^\dagger$ | $384^2$ | None | 2.9 M | 2.3 G | 77.0 |
| R5 | MobileViTv2-1.00 | $256^2$ | None | 4.9 M | 1.8 G | 78.1 |
| R6 | MobileViTv2-1.00$^\dagger$ | $384^2$ | None | 4.9 M | 4.1 G | 79.7 |
| R7 | MobileViTv2-1.25 | $256^2$ | None | 7.5 M | 2.8 G | 79.6 |
| R8 | MobileViTv2-1.25$^\dagger$ | $384^2$ | None | 7.5 M | 6.3 G | 80.9 |
| R9 | MobileViTv2-1.50 | $256^2$ | None | 10.6 M | 4.0 G | 80.4 |
| R10 | MobileViTv2-1.50 | $256^2$ | ImageNet-21k-P | 10.6 M | 4.0 G | 81.5 |
| R11 | MobileViTv2-1.50$^\dagger$ | $384^2$ | None | 10.6 M | 9.1 G | 81.5 |
| R12 | MobileViTv2-1.50$^\dagger$ | $384^2$ | ImageNet-21k-P | 10.6 M | 9.1 G | 82.6 |
| R13 | MobileViTv2-1.75 | $256^2$ | None | 14.3 M | 5.5 G | 80.8 |
| R14 | MobileViTv2-1.75 | $256^2$ | ImageNet-21k-P | 14.3 M | 5.5 G | 81.9 |
| R15 | MobileViTv2-1.75$^\dagger$ | $384^2$ | None | 14.3 M | 12.3 G | 82.0 |
| R16 | MobileViTv2-1.75$^\dagger$ | $384^2$ | ImageNet-21k-P | 14.3 M | 12.3 G | 82.9 |
| R17 | MobileViTv2-2.00 | $256^2$ | None | 18.5 M | 7.2 G | 81.2 |
| R18 | MobileViTv2-1.75 | $256^2$ | ImageNet-21k-P | 18.5 M | 7.2 G | 82.4 |
| R19 | MobileViTv2-2.00$^\dagger$ | $384^2$ | None | 18.5 M | 16.1 G | 82.2 |
| R20 | MobileViTv2-1.50$^\dagger$ | $384^2$ | ImageNet-21k-P | 18.5 M | 16.1 G | 83.4 |

Table 8: Performance of `MobileViTv2` on the ImageNet-21k-P validation set.

| Width factor $\alpha$ | # Params ↓ | FLOPs ↓ | Top-1 ↑ | Top-5 ↑ |
|---|---|---|---|---|
| 1.50 | 17.9 M | 4.1 G | 44.5 | 74.5 |
| 1.75 | 22.7 M | 5.5 G | 45.8 | 75.8 |
| 2.00 | 28.1 M | 7.2 G | 46.4 | 76.6 |

ImageNet-21k-P are able to achieve the similar performance with fewer FLOPs to models finetuned on ImageNet-1k with a higher resolution (e.g., R10 vs. R11; R14 vs. R15; R18 vs. R19 in Table 7).

**ImageNet-21k-P** Table 8 shows the results on the ImageNet-21k-P validation dataset. The performance of `MobileViTv2` improves with increase in model size.

## C  Comparisons with light-weight networks on the ImageNet-1k dataset

**Comparison with light-weight CNNs.** Fig. 7a shows that `MobileViTv2` outperforms light-weight CNNs across different network sizes (MobileNetv1 (Howard et al., 2017), MobileNetv2 (Sandler et al., 2018), ShuffleNetv2 (Ma et al., 2018), ESPNetv2 (Mehta et al., 2019), and MobileNetv3 (Howard et al., 2019)).

**Comparison with light-weight ViTs.** Fig. 7b shows that `MobileViTv2` achieves better performance than previous light-weight ViT-based models acorss different network sizes (DeIT (Touvron et al., 2021), T2T (Yuan et al., 2021), CrossViT (Chen et al., 2021a), LocalViT (Li et al., 2021), ConViT (d'Ascoli et al., 2021), and Mobile-former (Chen et al., 2021b)).

## D  Visualizations of separable self-attention scores

The `MobileViTv2` block, Fig. 6, unfolds the input $\mathbf{x} \in \mathbb{R}^{d \times H \times W}$ to obtain $\mathbf{x_u} \in \mathbb{R}^{d \times M \times N}$, where $N = \frac{HW}{hw}$ are the number of patches, each patch with width $w$ and height $h$ ($M = hw$ pixels per patch). This unfolded feature map is fed to separable self-attention module to learn non-local representations. To better understand how separable self-attention processes $\mathbf{x_u}$, we visualize context scores $\mathbf{c_s}$.

The separable self-attention in `MobileViTv2` block computes context scores $\mathbf{c_s}$ for $M$ pixels simultaneously across $N$ patches. Therefore, $\mathbf{c_s}$ has a dimensions of $M \times N$. To visualize context scores, we fold $\mathbf{c_s} \in \mathbb{R}^{M \times N}$ to

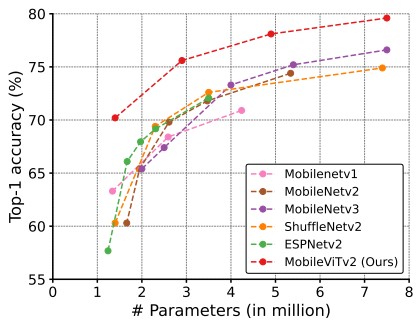
(a) Comparison with light-weight CNNs

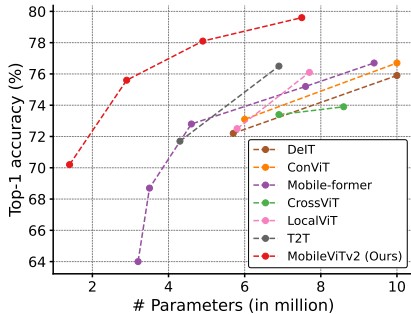
(b) Comparison with light-weight ViTs

Figure 7: **Comparison with light-weight CNN- and ViT-based models.** `MobileViTv2` is smaller and better, which is desirable for mobile devices.

the same spatial dimensions as the input and obtain context score map $\mathbf{c_m} \in \mathbb{R}^{H \times W}$. For ease of visualization, we scale $\mathbf{c_m}$ using min-max normalization.

The context score maps for different input images at different output strides of `MobileViTv2` model are shown in Fig. 8. These visualizations show that the proposed separable self-attention method is able to (1) aggregate information from entire image under different settings, including complex backgrounds, illumination & view-point changes, and different objects, and (2) learn high-, mid-, and low-level representations.

# E   `MobileViTv2`'s ablation studies on the ImageNet-1k dataset

In this section, we study the effect on different methods on the performance of `MobileViTv2` models, including augmentation methods.

**Standard vs. advanced augmentation**   We study two different augmentation methods: (1) standard augmentation that uses Inception-style augmentation Szegedy et al. (2015), i.e., random resized cropping and horizontal flipping and (2) advanced augmentation that uses RandAugment (Cubuk et al., 2020), CutMix (Yun et al., 2019), MixUp (Zhang et al., 2017), and RandomErase (Zhong et al., 2020) along with standard augmentation methods. The effect of these augmentations on the performance of `MobileViTv2` is shown in Figure 9. Smaller models ($< 4.5$ million parameters) benefit from standard augmentation while larger models ($\geq 4.5$ million parameters) benefit from advanced augmentation. For simplicity, we use advanced augmentation for all variants of `MobileViTv2` in this paper.

**Loss functions**   CutMix and Mixup augmentations mixes the samples in a batch. As a result, each sample has multiple labels. Therefore, in presence of these augmentations, ImageNet classification can be thought as a multi-label classification task. Similar to Wightman et al. (2021), we trained `MobileViTv2` by minimizing binary cross-entropy loss. Unlike Wightman et al. (2021), we did not observe any improvements in the performance when cross-entropy loss with label smoothing is replaced with binary cross-entropy loss. Therefore, we use cross-entropy with label smoothing for training `MobileViTv2` models.

**Effect of multiple latent tokens**   Similar to multi-head attention in transformers, the proposed separable self-attention can have multiple latent tokens. When we changed the number of latent tokens from 1 to 8, the performance improvements on the ImageNet-1k dataset were negligible (within $\pm 0.1$ top-1 accuracy). Therefore, we use only one latent token in our experiments.

We note that changing the number of heads from 4 to 1 in multi-headed self-attention in the transformer block of the MobileViTv1-S architecture dropped the top-1 accuracy by 0.7%. This observation is similar to Vaswani et al. (2017), who also found that multiple heads in multi-headed self-attention improve transformers performance on the task of neural machine translation.

**Improving FLOP-efficiency via pixel- and patch-sampling**  The MobileViTv1 model Mehta & Rastegari (2022) unfolds an input feature map into $N$ patches, each patch with $M = hw$ pixels and applies a transformer block for each pixel in a patch independently, where $h$ and $w$ are patch's height and width respectively. Because pixels in a patch are spatially correlated, one can sub-sample $m$ pixels from $M$ pixels and learn non-local representations by applying self-attention layers on $m$ pixels only. Such sub-sampling methods should help in reducing model FLOPs.

We tried following sampling methods at pixel- as well as patch-level:

- **Random sampling**, wherein $m$ pixels (or $n$ patches) from $M$ pixels (or $N$ patches) are randomly selected during training and uniformly during validation.

- **Top-$m$ (or top-$n$) sampling**, wherein top-$m$ pixels (or top-$n$ patches) are selected based on their magnitude computed using L2 norm.

- **Uniform sampling**, wherein $m$ pixels (or $n$ patches) are sampled uniformly from $M$ pixels (or $N$ patches).

We found that these methods can reduce the FLOPs by $1.2\times$ to $1.6\times$ with little or no drop in top-1 accuracy on the ImageNet-1k dataset for both MobileViTv1 (with multi-headed self-attention) and `MobileViTv2` (with the proposed separable self-attention) models. However, these improvements in FLOPs did not translate to latency improvements on a mobile device. In fact, models with these sampling methods were significantly slower than the models without these methods. The high-latency of models with these sampling methods on mobile devices can be attributed to their high memory access cost, as these methods change the memory order of tensor. Because of their high-latency on mobile devices, we did not use these methods in the `MobileViTv2` model.

## F  `MobileViTv2` training configurations

Configurations for training and finetuning `MobileViTv2`-2.0 on the ImageNet-1k and ImageNet-21k-P datasets are given in Table 9 and Table 10 respectively while configurations for finetuning `MobileViTv2` on downstream tasks are given in Table 11.

| Training config | MobileViTv2-2.0 | |
|---|---|---|
| Dataset | ImageNet-1k | ImageNet-21k-P |
| # Training samples | 1.28 M | 11 M |
| # Validation samples | 50 k | 523 k$^\dagger$ |
| Train resolution | $256 \times 256$ | $256 \times 256$ |
| Val resolution | $256 \times 256$ | $256 \times 256$ |
| RandAug | ✓ | ✓ |
| CutMix | ✓ | ✓ |
| MixUp | ✓ | ✓ |
| Random resized crop | ✓ | ✓ |
| Random horizontal flip | ✓ | ✓ |
| Random erase | ✓ | ✓ |
| Stochastic depth | ✗ | ✗ |
| Label smoothing | ✓ | ✓ |
| Loss | CE | CE |
| Optimizer | AdamW | AdamW |
| Weight decay | 0.05 | 0.05 |
| Scheduler | Cosine | Cosine |
| Warm-up iterations | 20 k | None |
| Warm-up init LR | $1e^{-6}$ | None |
| Warm-up scheduler | Linear | None |
| Base LR | 0.002 | 0.0003 |
| Epochs | 300 | 80 |
| Batch size | 1024 | 4096 |
| Layer-wise LR decay | ✗ | ✗ |
| Grad. clip | 10 | 10 |
| Exp. moving average | ✓ | ✓ |
| Weight init | Random | ImageNet-1k |

Table 9: Configuration for training MobileViTv2-2.0 on the ImageNet-1k/22k-P datasets. $^\dagger$ The validation set in ImageNet-21k-P does not overlap with ImageNet-1k validation set, and is created following Ridnik et al. (2021).

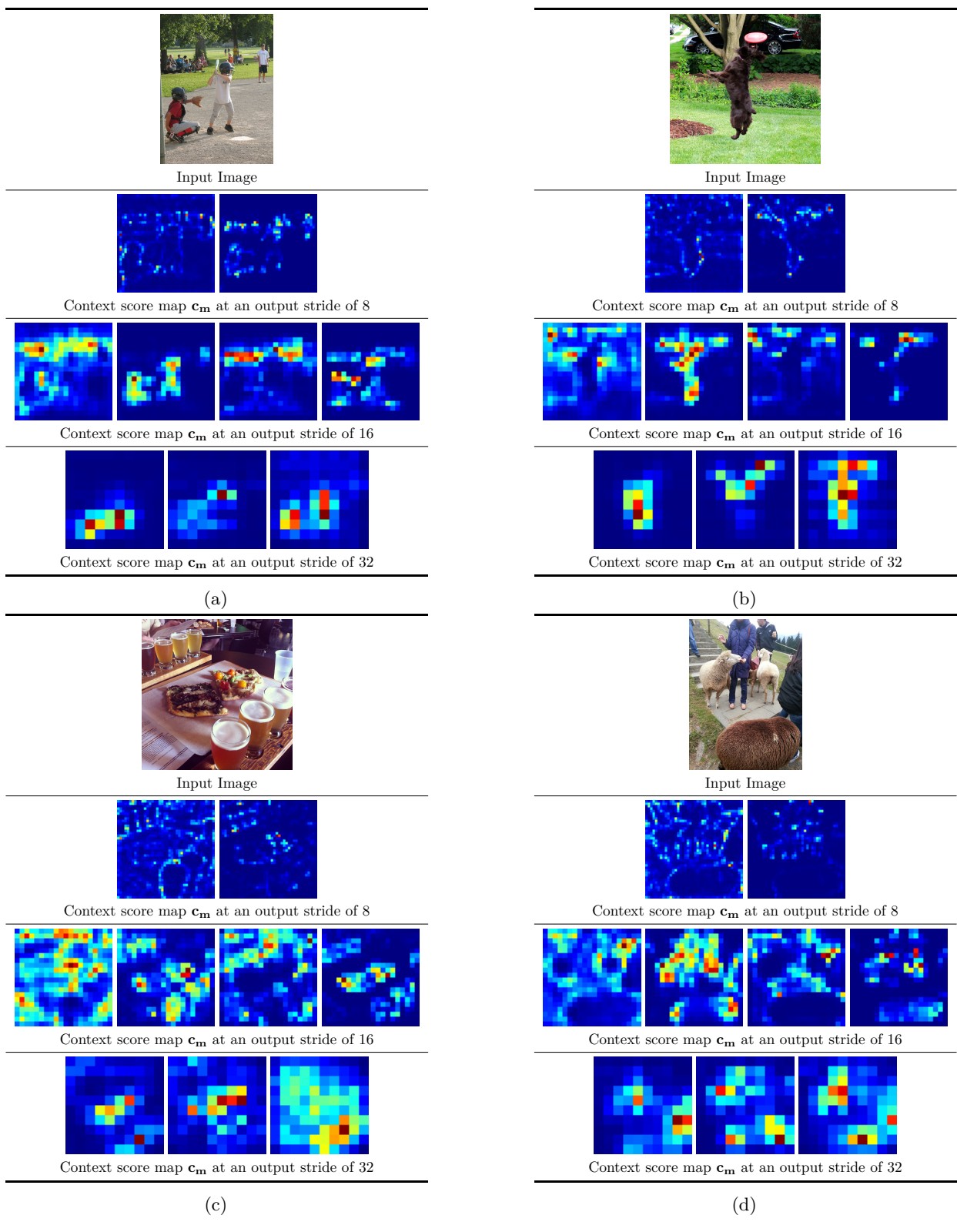

Figure 8: Layer-wise visualization of context score maps $\mathbf{c_m}$ at different output strides. Recall that MobileViTv2 (Fig. 6 and Table 6) applies $B = 2$, $B = 4$, and $B = 3$ separable self-attention layers at an output strides of 8, 16, and 32 respectively. Therefore, we have 2, 4, and 8 context score maps at an output stride of 8, 16, and 32 respectively

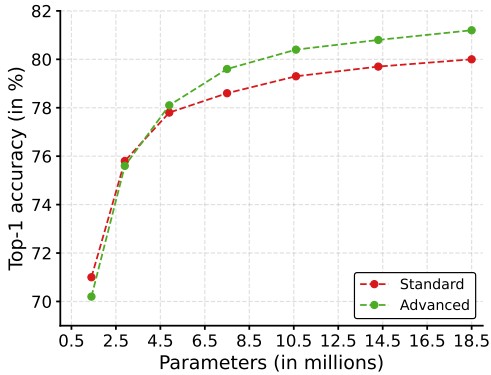

Figure 9: **Impact of data augmentation on the performance of `MobileViTv2` models on the ImageNet-1k dataset.** For smaller models ($< 4.5$ million parameters), standard augmentation works best while larger models ($\geq 4.5$ million parameters) benefit from advanced augmentation.

| Training config | MobileViTv2-2.0 | | |
|---|---|---|---|
| Dataset | ImageNet-1k | ImageNet-1k | ImageNet-1k |
| # Training samples | 1.28 M | 1.28 M | 1.28 M |
| # Validation samples | 50 k | 50 k | 50 k |
| Train resolution | $384 \times 384$ | $256 \times 256$ | $384 \times 384$ |
| Val resolution | $384 \times 384$ | $256 \times 256$ | $384 \times 384$ |
| Weight init | ImageNet-1k | ImageNet-21k-P | ImageNet-21k-P-1k$^\dagger$ |
| RandAug | ✗ | ✓ | ✓ |
| CutMix | ✗ | ✓ | ✓ |
| MixUp | ✗ | ✓ | ✓ |
| Random resized crop | ✓ | ✓ | ✓ |
| Random horizontal flip | ✓ | ✓ | ✓ |
| Random erase | ✗ | ✓ | ✓ |
| Stochastic depth | ✗ | ✗ | ✗ |
| Label smoothing | ✓ | ✓ | ✓ |
| Loss | CE | CE | CE |
| Optimizer | SGD | SGD | SGD |
| Weight decay | $4e^{-5}$ | $4e^{-5}$ | $4e^{-5}$ |
| Scheduler | Fixed | Cosine | Fixed |
| Warm-up iterations | None | None | None |
| Warm-up init LR | None | None | None |
| Warm-up scheduler | None | None | None |
| Base LR | 0.001 | 0.01 | 0.001 |
| Epochs | 10 | 50 | 10 |
| Batch size | 128 | 256 | 128 |
| Layer-wise LR decay | ✗ | ✗ | ✗ |
| Grad. clip | 10 | 10 | 10 |
| Exp. moving average | ✓ | ✓ | ✓ |

Table 10: Configuration for finetuning `MobileViTv2-2.0` on the ImageNet-1k dataset. Here, $^\dagger$ denotes that the ImageNet-21k-P model finetuned on the ImageNet-1k dataset at $256 \times 256$ image resolution is used for initializing the weights.

| Training config | SSDLite-MobileViTv2-1.75 | DeepLabv3-MobileViTv2-1.75 | DeepLabv3-MobileViTv2-1.75 |
|---|---|---|---|
| Dataset | MS-COCO | ADE20k | PASCAL VOC 2012 |
| Extra Data | None | None | COCO |
| Task | Detection | Segmentation | Segmentation |
| # Training samples | 117 k | 20 k | 128 k |
| # Validation samples | 5 k | 2 k | 1.45 k |
| Train resolution | $320 \times 320$ | $512 \times 512$ | $512 \times 512$ |
| Val resolution | $320 \times 320$ | Shortest side 512 | Shortest side 512 |
| Weight init | ImageNet-1k | ImageNet-1k | ImageNet-1k |
| SSD Cropping | ✓ | ✗ | ✗ |
| Photometric distortion | ✓ | ✓ | ✓ |
| Random horizontal flip | ✓ | ✓ | ✓ |
| Resize | ✓ | ✗ | ✗ |
| Random short size resize | ✗ | ✓ | ✓ |
| Random Crop | ✗ | ✓ | ✓ |
| Random Gaussian blur | ✗ | ✓ | ✓ |
| Random rotation | ✗ | ✓ | ✓ |
| Loss | Smooth L1 + CE | CE | CE |
| Optimizer | AdamW | SGD | AdamW |
| Weight decay | 0.05 | $1e^{-4}$ | 0.05 |
| Scheduler | Cosine | Cosine | Cosine |
| Warm-up iterations | 500 | None | 500 |
| Warm-up init LR | $9e^{-5}$ | None | $5e^{-5}$ |
| Warm-up scheduler | Linear | None | Linear |
| Base LR | 0.0009 | 0.02 | 0.0005 |
| Epochs | 200 | 120 | 50 |
| Batch size | 128 | 16 | 128 |
| Layer-wise LR decay | ✗ | ✗ | ✗ |
| Grad. clip | 10 | 10 | 10 |
| Exp. moving average | ✓ | ✓ | ✓ |

Table 11: Configuration for finetuning `MobileViTv2` on downstream tasks. For Ade20k, we found that SGD was more stable as compared to AdamW across different `MobileViTv2` configurations, and therefore, we used SGD for finetuning on Ade20k dataset. The configurations for `MobileViTv2` with PSPNet are the same as Deeplabv3 on both PASCAL VOC and Ade20k datasets.

