# OpenReview forum: "Separable Self-attention for Mobile Vision Transformers"
_TMLR — Accepted by TMLR_

### Review · Reviewer_fE6c · 2022-10-25

**Summary Of Contributions:**

This work propose separable self-attention for mobile vision transformers. Specifically, it first profile different transformers model on mobile devices and detect multi-head self-attention as the one the bottleneck. Then it proposes a separable self-attention scheme to convert the batched matrix multiplication in standard multi-head self-attention into element-wise operations, thus the complexity decreases from O(k^2) to O(k), with k as the sequence length. The experiments on both ImageNet and MS-COCO verify that the new MobileViTv2 equipped with the proposed separable self-attention can achieve better accuracy vs. mobile device latency trade-offs as compared to MobileViT and some other efficient transformer baselines.

**Audience:**

Yes

**Broader Impact Concerns:**

No concern on the ethical implications.

**Claims And Evidence:**

Yes

**Requested Changes:**

It would be better if the authors can address the Weaknesses mentioned above, the three points are critical to securing my recommendation for acceptance.

**Strengths And Weaknesses:**

## Strengths
> + Clear logic flow: the paper is well-written. I really like their writing style of starting from analysis on the real devices, which provides solid motivation to the proposed separable self-attention.
> + Discuss the failure cases: although many papers are trying to hide unsatisfying, this work directly shows that there is still a gap between ViT and Conv-based models in terms of the accuracy vs. mobile device latency even for their proposed MobileViTv2. It is also a good point to show to the community and motivate more works on optimization ViT latency on mobile devices.
> + Comprehensive experiments: the benefit of the proposed MobileViTv2 is supported by experiments on different datasets and applications, include image classification, semantic segmentation, and object detection.

## Weaknesses
> + Not cover enough baseline for comparison: The proposed separable self-attention still belongs to the linear attention category but only Linformer is included in the comparison, more linear self-attention works should be included in the comparison.
> + Need more evidence for "The main efficiency bottleneck in MobileViT is the multi-headed self-attention
(MHA) in transformers": as shown in Figure 1, BMM corresponding to self-attention is not the most costly operator but the MM corresponding to linear layers is. So the claim may need modifications or more evidences.
> + Need more devices to support the claimed improvement on mobile devices: In experiments, only iPhone 12 is used as the devices to run benchmark.  However, it is very powerful smartphone with mobile GPU. It would be better to add the performance on some other devices, especially those phones without any specific accelerator.

---

### Review · Reviewer_uYAm · 2022-10-26

**Summary Of Contributions:**

This paper improved the existed mobile vit by replacing the multi-head self-attention by the proposed separable self-attention. The paper show the results in many visual tasks.


**Audience:**

Yes

**Broader Impact Concerns:**

This paper proposed new building block for a neural network, as others did, it shares many of the pitfalls and the benefits associated with deep learning models.


**Claims And Evidence:**

Yes

**Requested Changes:**

Please see the pointed listed in weakness sections.

And there are a few more points:

1. In the visualization, the authors should also show the result of MobileViTv1 for the comparison.

2. In Fig. 1, I think using the analysis of mobile device, e.g. iPhone 12, will make the story more consistent. As the architecture of CPU is also different from mobile device.

3. Missing citation, as I think the concept of the paper is similar to LambdaNet [1], which also model the global context directly instead of multi-head self-attention.

[1] LAMBDANETWORKS: MODELING LONG-RANGE INTERACTIONS WITHOUT ATTENTION, ICLR 2021


**Strengths And Weaknesses:**

Strengths:
1. The paper analyzed the latency of few represented models and pointed out that some specific operation is not suitable for mobile device and then improve based on that.
2. The proposed module is simple and effective and the authors evaluated on many visual tasks.

Weaknesses:
1. When comparing the ConvNeXt, the proposed MobileViTv2 only outperform it in terms of parameters. Even though the authors mentioned the ViT-based models exhibit better scaling properties but it seems that it conflicts with the goal of the paper, which aims for resource-constrained device. Moreover, if we compare R4 and R13, the ConvNeXt-T provided similar throughput and latency but much better accuracy. I think the authors should elaborate more on this.

2. Figure 2 shows the comparison with MobileViTv1 but why there is no comparison to MobileViTv1 in Table 3 for the ADE20k dataset and there is no latency comparison for both Table 3 and 4.

3. In Table 3 and 4, why not compare to ConvNeXt as well as MobileViTv2-2.0 also listed in the table?

---

### Review · Reviewer_AaHu · 2022-11-21

**Summary Of Contributions:**

The paper presents an efficient self-attention mechanism namely separable self attention to improve the latency of transformer models for resource constrained applications. In particular, the authors have proposed to replace batch-wise matmul with context aware element-wise operations. Results on various FLOPS/Param budget show the efficacy of the proposed method.

**Audience:**

Yes

**Broader Impact Concerns:**

The broader impact section is not provided in the current draft.

**Claims And Evidence:**

Yes

**Requested Changes:**

Please refer to the weakness.

**Strengths And Weaknesses:**

Strengths:
-----------------
* The paper is well motivated with impressively written introduction and updated related works.
* The writing is good with good and thorough empirical evaluations to defend the claims made in the paper.
* Highlight of the latency issue of Linformer with supporting results.
* The Fig. 3 and 4 are well explanatory.
* Results with downstream tasks are inspiring.

Weaknesses:
-----------------
* Please provide additional comparisons with EfficientFormer [1].
* Please add CPU latency comparison with MobileViT in the Fig. 1 bottom right.
* The mention of optimization (pruning) and distillation is provided as two separate strategy, which is the general case. However, recent research has provided joint optimization to perform both pruning and distillation via a single training loop [2]. Thus, this part needs to be updated with mention of such methods to provide a wholesome scope of the possible options.
* Also, the authors should touch upon various NAS ViT methods that can yield accuracy-complexity pareto front suitable for target hardware. In particular, it would be interesting to see a discussion (and possibly results) of the current model in the context of NAS.
*  The code and/or training script is not provided for reproducibility check. As EfficientFormer can be treated as aform of NAS, a detailed comparison with that would do as well (in understanding the value of the current separable self-attention method).
* Please provide comparison with other ViT variants on downstream tasks apart from providing comparison with only MobileViT-1.
* Please show the default robustness of this model variants. Generally, ViTs are inherently robust against OOD tasks. **It is important to understand whether such separable self-attention modules impact on that or not**.

[1] EfficientFormer: Vision Transformers at MobileNet Speed, NeurIPS 2022.

[2] AttentionLite: Towards Efficient Self-Attention Models for Vision, ICASSP 2021.

---

### Decision · Action_Editors · 2023-01-09

**Recommendation:** Accept with minor revision

**Comment:**

The reviewers have all found the paper to be well written, the experiments strong and well motivated. All three reviewers have recommended acceptance, subject to a number of relatively minor changes that I believe the authors should be able to achieve: such as updating the related works with the citations pointed out by reviewers in their initial reviews, adding the rebuttal & discussion results into the main paper as well as making a bona-fide effort for open-sourcing this work. If the latter is there, I'd vote for having this certified as reproducible.

Some specific recommendations also include having (1) more evidence and discussion for "The main efficiency bottleneck in MobileViT is the multi-headed self-attention (MHA) in transformers" in the paper itself and (2) more baseline comparisons in addition to Linformer (+discussion for why some comparisons may be apples to oranges).

**Audience:**

I believe anyone deploying self-attention to mobile devices where latency is crucial would benefit from the findings of this work. This represents a healthy chunk of the TMLR audience.

**Claims And Evidence:**

This work investigates modifying self-attention in a way that replaces batch multiplication with element-wise operations. This reduces complexity in a very meaningful fashion (to linear). The main finding is that this is a good way to reduce latency, especially for mobile devices. The work proposes a simple and effective module that was evaluated on a number of visual tasks such as ImageNet and MS-COCO. The proposed MobileViTv2 compares favorably in terms of accuracy while being significantly faster.